# Continuity of Care and Self-Management among Patients with Stroke: A Cross-Sectional Study

**DOI:** 10.3390/healthcare9080989

**Published:** 2021-08-04

**Authors:** Nai-Yu Kuo, Yu-Huei Lin, Hsiao-Mei Chen

**Affiliations:** 1Department of Nursing, Chung Shan Medical University, Taichung 40201, Taiwan; 101449@cch.org.tw; 2Department of Nursing, Changhua Christian Hospital, Changhua 500024, Taiwan; 3Post-Baccalaureate Program in Nursing, Taipei Medical University, Taipei 11031, Taiwan; gracelin@tmu.edu.tw; 4Department of Nursing, Chung Shan Medical University Hospital, Taichung 40201, Taiwan

**Keywords:** stroke, continuity of care, self-management

## Abstract

(1) Background: Stroke is an important topic in the healthcare industry. The objective of the present study was to investigate patients’ sociodemographic characteristics, health status, continuity of care, self-management, and other predictors that affect their self-management. (2) Methods: This cross-sectional correlational study was carried out from March to September 2020, and included a total of 150 patients aged 20 and above who were diagnosed within the past 6 months. The research participants were selected from the Division of Neurology, Department of Internal Medicine/Department of Surgery, at a medical center in Central Taiwan. (3) Results: The mean self-management score of patients with stroke was 110.50 points (30–150 points). As shown in the stepwise regression analysis, the overall regression model explained approximately 44.5% of the variance in self-management. Educational level (10.8%), frequency of exercise per week (2.1%), time that patients were affected by stroke (2.4%), and continuity of care (29.2%) were the main predictors affecting the self-management of stroke patients. (4) Conclusions: To improve stroke patients’ self-management, medical teams should provide appropriate continuity of care to those with lower educational levels, those without exercise habits, and those who experienced a stroke within the past six months.

## 1. Introduction

Stroke is the main cause of long-term disability in patients [1]. According to the statistics, approximately 20% of patients with stroke either use wheelchairs or are bedridden. Meanwhile, more than 60% of patients discharged from hospital still experience neurological sequelae, such as skewed faces, slow and inflexible hands and feet, or unclear speech, which leads to a heavy burden on individuals, families, and society [2]. The activities of daily living (ADLs) of stroke patients may be regarded as an indicator of their level of life independence, which subsequently affects their self-management ability [3]. The level of dependence in patients’ ADLs can be evaluated using the modified Rankin Scale (mRS). The status of ADLs can be divided into 7 levels, from 0 to 6, and it must be noted that any change in these levels is clinically significant [4]. The National Institute of Health Stroke Scale (NIHSS) has also been widely used to evaluate stroke severity. A higher score on the scale implies more severe neurological damage in patients and is associated with self-management [5].

Continuity of care (COC) refers to a series of care activities designed for patients during hospitalization, including discharge planning, case management, and referral to an adequate subsequent care institution. A professional team conveys health information to patients throughout these care activities to achieve a good disease management outcome [6,7]. Hadjistavropoulos et al. [8] indicated that COC includes three major dimensions: information, relationship, and management. During the long-term care process, patients with stroke may be referred to a different healthcare institution, where the level of familiarity with care skills may differ widely. Therefore, the COC clinically provided by caregivers to patients should focus on the “relationships with providers in hospital” and the “information transfer to patients.” A well-designed discharge preparation service plan should also be provided before patients are discharged from the hospital. Cross-professional team cooperation is useful in integrating resources and assisting patients and their primary caregivers in follow-up care to obtain COC services [6].

Boger et al. [9] suggested that self-management is a strategy adopted by individuals in managing long-term health status and is potentially important for disease reduction and an individual’s healthcare. As proposed by Corbin and Strauss [10] and Lorig and Holmon [11], self-management includes three major scopes: medical management, role management, and emotional management. Through medical management, which is concerned with disease control, patients contact healthcare personnel for medical consultations in the event of a deterioration in their symptoms [12]. Been-Dahmen et al. [13] suggested that role management refers to patients’ ability to maintain a healthy lifestyle, which means that patients can incorporate regular healthy lifestyle choices and find suitable or supportive roles in their daily lives to help them to adapt to their health condition. Emotional management refers to patients’ ability to deal with the emotions (e.g., depression, anger, frustration, and guilt) caused by chronic disease and manage their interpersonal relationships with friends and family [14,15]. Relevant studies have shown that patients’ active participation in self-management promotes recovery and improves health outcomes. However, first-time stroke patients are not familiar with the relevant information about nursing care and rehabilitation plans, and their management and skills of ADLs are severely insufficient, generating many problems in the care of patients transitioning from the hospital to the home [16], which might have resulted from patients’ insufficient access to COC, limitations of services, and being provided with insufficient information [7,16,17]. Therefore, promoting self-management among patients for stroke prevention and improving health outcomes is essential [7,18].

Self-management has been widely applied to chronic illnesses, among which the most common are diabetes [19] and cardiovascular diseases [20]. In addition, studies have indicated that self-management interventions could enable patients to perform effective management of their health and change their attitudes and behavior [9]. It has also been found, in a systematic review and meta-analysis of self-management in stroke patients, that self-management could improve the quality of life, lifestyle, and drug compliance among stroke patients living in the community [21,22]. However, there are still limited research findings on the COC and self-management of hospitalized first-time stroke patients. Therefore, the research purposes of this study are: (1) to understand the current data on the basic demographics, health status, COC, and self-management of patients with stroke, (2) to investigate the correlation among patients’ basic demographic data, health status, COC, and self-management status, and (3) to investigate the important predictors for self-management. The findings can serve as a reference for professional clinical teams in terms of enabling patients to receive COC and further improve their self-management.

## 2. Materials and Methods

### 2.1. Study Design and Participants

This descriptive, correlational, cross-sectional study used convenience sampling to enroll patients with stroke from the wards of the Division of Neurology, Department of Internal Medicine/Department of Surgery at a medical center in central Taiwan as research participants. The inclusion criteria for the research participants were as follows: (1) over the age of 20 who were newly diagnosed with stroke within the past six months, (2) mRS grades 0–3, and (3) agreed to participate in this study and signed the informed consent form. Patients with severe dementia, acute respiratory syndrome, or visual or hearing impairments were excluded from the study. Furthermore, this study collected data from 1 March to 30 September 2020.

This research study used statistical software G power 3.1.9.2 [23] to estimate the sample size. According to Cohen [24], the statistical power value = 0.8, the effect size *f*^2^ = 0.19, α = 0.05, the number of independent variables = 19, and *R*^2^ = 0.16; thus, the sample size should be 124. However, considering that some participants may fail to complete all the questionnaires, this study increased the research sample size by 20% [25] and enrolled 150 participants in total, with zero invalid questionnaires and a 100% participation rate.

### 2.2. Measurement

The tools used in this study included scales on basic demographic data, health status, COC, and self-management. The basic demographic data included age, gender, marital status, educational level, economic status, and religious belief.

The health status scale collected information on smoking status, exercise frequency, time affected by stroke, total number of chronic diseases, ADLs, and instrumental activities of daily living (IADLs). Moreover, patients’ NIHSS and mRS data were collected from medical charts. Both ADLs and IADLs, with a total score of 100 points, were developed by Mahoney and Barthel in 1965 [26]. This scale contains 10 assessment items, including eating, moving, personal modification, and toileting ability, and primarily measures stroke patients’ level of life dependence. In particular, the IADL scale (total score: 24 points), developed by Lawton and Brody [27], was used to evaluate the research participants’ ability to maintain their independence and autonomy, such as cooking, shopping, and making phone calls. The Cronbach’s α value of ADLs and IADLs in this study was 0.92 and 0.90, respectively. The NIHSS, an assessment scale for neurological deficits in patients with acute stroke, has a total score of 42 points. More than 80% of patients with NIHSS scores lower than or equal to 5 points, who experienced mild neurological deficits, can directly return home after being discharged from the hospital. Most patients with an NIHSS score of 6–13 points need to be hospitalized for rehabilitation, while those with NIHSS scores greater than 14 points usually need long-term care [28]. The Cronbach’s α value of the NIHSS in this study was 0.83. The modified Rankin Scale (mRS) was used to evaluate the grading of neurological impairment and the patients’ level of disability. The assessment results of this scale divide the functional status of patients into seven grades (grades 0–6). A higher grade indicates a higher level of disability, according to Maredza et al. [4]. Relevant studies have shown that the reliability of mRS is as high as 0.94, while its test–retest reliability is 0.81–0.95. In addition, its construct validity is also significant [29]. The Cronbach’s α value of mRS in this study was 0.95.

The traditional Chinese COC scale (Patient Continuity of Care Questionnaire, PCCQ) was developed by Chen and Chen [6], based on the translation of the PCCQ scale of Hadjistavropoulos et al. [8]. The PCCQ scale was chosen for this study because it is a localized COC questionnaire with good reliability and validity, and is applicable for patients with multiple chronic diseases [6]. The content of the COC scale, which comprises 12 items, concerns two variables, namely, the “relationships with providers in hospital” (five items) and the “information transfer to patients” (seven items) (see Appendix A). This scale uses a five-point Likert scale for scoring, with a total score ranging from 12 to 60 points. According to the reliability and validity of the COC scale for patients with chronic diseases, tested by Chen and Chen [6], the internal consistency (Cronbach’s α value) of the scale was 0.92, the test–retest reliability was 0.94, and the content validity index (CVI) was 0.93. Therefore, the reliability and validity of the scale were good. The Cronbach’s α value of each of the overall scales of COC, “relationships with providers in hospital” and “information transfer to patients,” in this study was 0.98.

The Stroke Self-Management Scale was developed by the researchers based on the scale contents of four previous studies, namely Lin et al. [30] (2011), Song and Lin [31], Boger et al. [9], and Battersby et al. [32]. The main reason for developing the scale was because that there is no suitable and localized stroke self-management scale in Taiwan. Attempts were made to obtain the authorization to use the Southampton Stroke Self-Management Questionnaire (SSSMQ) developed by Boger et al. [9], but were not successful. Therefore, the researchers referred to the existing self-management scales for patients with stroke, diabetes, hemodialysis, and chronic conditions [9,30,31,32], and modified the contents into the self-management scale for this study. This scale was tested for reliability and validity, and was administered to evaluate the self-management of first-time stroke patients. The scale contains 30 items, and is divided into 3 parts, namely, medical management (13 items), role management (12 items), and emotional management (5 items). A five-point Likert scale was used for scoring (see Appendix A), and the mean score of each scale dimension was 30–150 points. This study further used an expert validity test to evaluate the validity of each item of the scale. The content validity index (CVI) value was 0.96, while the surface validity (dichotomy) was 0.95. Moreover, the Cronbach’s α value of the overall self-management scale was 0.93. In particular, the α value of the medical self-management was 0.91, that of the role of self-management was 0.90, and that of the emotional self-management was 0.68.

### 2.3. Data Collection

This study was approved by the Institutional Review Board (IRB) of the selected research site (approval number, 191235) before its initiation and was deemed compliant with the moral guidelines of the Declaration of Helsinki, established in 1975. Consent from the hospital was obtained to enroll participants from the wards of the Division of Neurology, Department of Internal Medicine/Department of Surgery. The hospital’s nursing supervisor assisted in determining the in-patients who met the inclusion criteria: patients over the age of 20 diagnosed within the past six months. Moreover, the researchers duly explained the purpose of this study to the enrolled participants and obtained their consent. The participants were then requested to fill out the informed consent form and the scales, which took approximately 20–30 min each to complete. Most notably, the research participants were informed that they could immediately withdraw from the study at any time without any reason; 150 patients were enrolled, with no withdrawal or loss of patients throughout the study.

### 2.4. Data Analysis

After the data coding, this study used the SPSS 25.0 (IBM Corp., Armonk, NY, USA) software package in Chinese for descriptive statistical analysis, such as frequency, percentage, mean, and standard deviation. Standardized score conversion was performed on the overall scales and subscales of the Continuity of Care Scale and the Stroke Self-Management Scale to convert the original scores into the standard score of 100. In addition, this study used the independent samples *t*-test, Pearson’s product–moment correlation, one-way ANOVA, and multiple stepwise regression analyses for inferential statistical analysis. Furthermore, this study used the Kolmogorov–Smirnov (KS) test to verify whether the residual items of the regression model conformed to normal distribution. For all the statistical analyses, a *p*-value of <0.05 indicated that the estimate of the variable had statistical significance.

## 3. Results

### 3.1. General Status of the Basic Demographic Data, Health Status, Continuity of Care, and Self-Management of Patients with Stroke

This study investigated a total of 150 stroke patients with a mean age of 65.15 ± 13.62 years. Out of the 150 participants, 84 (54.0%) were ≥65 years of age, 107 (71.3%) were male, 68 (45.3%) graduated from junior to senior high school, and 76 (50.7%) had salary as the main source of income. In terms of health status, the average total number of diseases affecting the participants was 2.35 ± 1.35 diseases. According to the diagnoses by physicians, the three most prevalent chronic diseases affecting the participants were in the following order: 99 participants (66.0%) experienced hypertension, 48 participants (32.0%) experienced hyperlipidemia, and 44 participants (29.3%) experienced diabetes. The time affected by stroke for 146 participants (97.3%) was within one month. Furthermore, in terms of exercise habits, 86 (57.3%) participants did not have the habit to exercise. The mean NIHSS of the participants was 1.89 ± 1.68, while the NIHSS of 124 participants (82.7%) was 0–3 points, suggesting mild stroke. The mean grade of mRS was 1.82 ± 1.00, while the mRS of 91 participants (60.7%) was grades 2–3. Moreover, the mean ADL was 87.10 ± 18.86 points, with a total score of 100 points, suggesting moderate dependence. The mean IADL was 19.85 ± 5.71 points, with a total score of 24 points (Table 1).

Additionally, the mean score of self-management in this study was 110.5 ± 15.12. The mean scores of the medical management, role management, and emotional management were 51.74 ± 6.80, 42.58 ± 7.93, and 16.18 ± 3.33, respectively. Among them, the score indicator of medical management was higher, at 79.60, while that of emotional management was the lowest, at 64.72 (Table 2). The mean score of PCCQ was 46.61 ± 8.99. Specifically, the “relationships with providers in hospital” generated a score of 19.46 ± 3.82, while the “information transfer to patients” generated a score of 27.14 ± 5.35 (Table 2).

### 3.2. Relationship between the Basic Demographic Data, Health Status, and Continuity of Care of Patients with Stroke and Self-Management

The basic demographic data of the patients (e.g., gender, marital status, religious belief, and sufficient financial support) were not significantly associated with the patients’ self-management. However, a significant correlation was rather found in all other variables. Age was negatively correlated (*p* < 0.01) with the overall self-management, medical self-management, and role of self-management of patients with stroke. In particular, self-management was better among younger patients. It should be noted that the patients’ educational level also had a significant correlation with the overall self-management, medical self-management, and role of self-management (*p* < 0.001). Patients with an educational background of junior college and above showed better self-management than those of (vocational) senior high school and elementary school. Additionally, the patients’ main source of income had a significant correlation with the overall self-management and the role of self-management (*p* < 0.05). Those patients who relied on salary income showed better self-management than those who relied on their children/spouse/siblings/parents for income (Table 3).

Moreover, this study found that the frequency of exercise per week, time affected by stroke, ADLs, IADLs, mRS grade, and NIHSS were significantly correlated with self-management in terms of health status. The exercise frequency of patients with stroke was found to have a significant correlation with their overall self-management (*p* < 0.05). The self-management of patients who exercised one to two times or more than three times a week was better than that of those never exercised. The patients’ time affected by stroke also had a significant correlation with self-management. The results showed that the overall self-management, medical self-management, and role of self-management of patients who experienced a stroke within two to six months were better than those who experienced a stroke within one month. Moreover, the ADLs and IADLs of patients were significantly positively correlated with the overall self-management, medical, and role of self-management (*p* < 0.01). This finding implies that the higher the ADL and IADL scores, the better patients’ self-management was. NIHSS, on the other hand, was significantly negatively correlated with the overall self-management (*p* < 0.05) and the role of self-management (*p* < 0.01). This finding suggests that the higher the NIHSS of patients (more severe stroke), the poorer their overall self-management and role of self-management were. Furthermore, the mRS grade was also significantly negatively correlated with the overall self-management, medical self-management, and role of self-management. A higher overall disability of patients implies poorer overall self-management, medical, and role of self-management (*p* < 0.01). Meanwhile, patients’ COC was significantly positively correlated with the overall self-management, medical self-management, and role of self-management (*p* < 0.01) (Table 3).

It is noteworthy that the “relationships with providers in hospital” and the “information transfer to patients” in the COC were also significantly positively correlated with the overall self-management, medical self-management, and role of self-management (*p* < 0.01). With better “relationships with providers in hospital” and more “information transfer to patients” in the COC, the patients’ self-management was better (Table 4).

### 3.3. Important Predictors Affecting the Self-Management of Patients with Stroke

This study used multiple stepwise regression analyses to determine the primary factors affecting stroke patients’ self-management. In addition, this study placed factors that were statistically significantly correlated with self-management into the regression model. For the stepwise linear regression analysis, this study further set the patients’ age, educational level, main source of income, exercise frequency, time affected by stroke, ADLs, IADLs, NIHSS, mRS, and COC as the independent variables, and their overall self-management as the dependent variable. Prior to performing the regression analysis, this study converted categorical variables, such as the educational level, main source of income, and exercise frequency, into dummy variables. Afterward, the variables were tested as to whether the regression model of self-management of patients with stroke, according to Kolmogorov–Smirnov (KS = 0.92, *p* > 0.05), had a normal distribution. Collinearity diagnostics of each independent variable were performed to determine the independence between each variable. The tolerance values between each variable were all greater than 0.10 (0.982–0.990), and the variance inflation factor (VIF) was less than 10 (1.011–1.018), indicating that there was no problem of collinearity among the independent variables.

The results of the analysis showed that the educational level, exercise frequency, time affected by stroke, and COC were the best predictive variables for self-management. COC had the greatest explanatory power, which could explain 29.2% of the variance. This factor was then followed by the junior college and above educational level (10.8%), experiencing a stroke within two to six months (2.4%), and exercise frequency of one to two times per week (2.1%). The variables mentioned above could effectively explain 44.5% of the variance of the overall patients’ self-management (*F* = 30.921, *p* < 0.001). The results showed that after controlling the basic demographic characteristics, health status, and COC of patients with stroke, there was a significant difference in self-management (as shown in Table 5).

## 4. Discussion

This study showed that demographic characteristics and health status, such as younger age, higher educational level, exercise frequency of one to two times per week, experiencing a stroke within two to six months, ADLs, IADLs, NIHSS, mRS, and COC, were significantly correlated with self-management. Furthermore, this study found that there was better self-management among younger patients. This result is notably consistent with the findings of Guan et al. [33]. Relevant studies have pointed out that stroke patients aged under 40 are more willing to perform health behaviors to restore their health (diet, medication, social interaction, interpersonal relationship, changes in daily lifestyle, emotions, rehabilitation, and illness treatment and prevention) [33,34]. Moreover, younger patients are more concerned about their health, proactively seek and value information provided by healthcare personnel, and engage in exercises to regain health. Therefore, self-management among younger patients is better than among the elderly [33]. The research results also showed that the self-management of patients with an educational level of junior college and above was significantly better than that of patients of elementary school, which could explain 10.8% of the variance. This result is consistent with the findings of Guan et al. [33], suggesting that patients with higher educational levels better understand disease self-management and proactively engage in meaningful self-management behaviors.

In terms of health status, the self-management of patients with stroke who exercise one to two times per week was better than those who never exercised, and it could explain 2.1% of the variance. Stroke patients who show a stronger intention to exercise appeared to have better self-management; therefore, professional medical teams should encourage the patients to participate in physiotherapy during hospitalization in order to strengthen their self-management abilities [35]. Additionally, it was found in this study that although there were only 4 patients diagnosed with their first stroke within 2–6 months, their scores in self-management, medical management, and role management were better than the other 146 patients who were diagnosed within 1 month, which could explain 2.4% of the variance. Furthermore, Ruksakulpiwat and Zhou [36] suggested that the information provided by medical teams to patients who suffered from stroke within 6–12 months could bridge gaps and meet their needs in proactively managing diseases. This is possibly because stroke survivors request more information and medical support from medical experts regarding stroke recovery due to disease progression [16,17]. Therefore, the self-management of patients with stroke improves over time after they suffered from the stroke [37].

This study also found that patients with higher ADL and IADL scores have better self-management. Spoorenberg et al. [38] suggested that an impairment in ADLs and IADLs leads to a decline in stroke patients’ self-management, which further affects self-management behaviors. Meanwhile, this study found that the NIHSS scores of patients was significantly negatively correlated with self-management. This result is particularly consistent with the results of Damush et al. [5], who suggested that patients with a lower severity in the NIHSS are more willing to engage in disease self-management. This study also found that the mRS grade of stroke patients is negatively correlated with self-management, which implies that there is poorer self-management among patients whose score was higher in the overall functional disability [5,38]. The characteristics of the samples of the study were associated with mild stroke, and the overall neurological functions indicated mild disability; however, they were moderately dependent in terms of their ADLs, such as toileting, dressing, and ambulating, similar to the characteristics of the participants in Hagberg et al. [39]. Therefore, it is suggested that patients, after their first stroke, should participate in the pilot program for acute post-stroke care organized by the Central Health Insurance Agency of the Ministry of Health and Welfare, and receive 6–12 weeks of integrated care for active and highly intense rehabilitation. Active rehabilitation during the primary period of treatment among stroke patients could reduce their disability level, and it would help stroke patients to improve their self-management [40].

It is noteworthy that this study included variables associated with the COC of patients. These variables helped to assess the influence of COC toward stroke patients’ self-management. The research results showed that patients’ COC is positively correlated with self-management and explains 29.2% of the variance. Moreover, this study found that there was better self-management among patients with better COC, which is consistent with the results of other qualitative research. Lack of disease care-related information during hospitalization and failure to establish relationships and communication between healthcare personnel and patients could hinder the latter’s implementation of self-management [41]. Therefore, medical teams must ensure and strengthen the COC among patients during hospitalization in order to convey sufficient personal health information and provide appropriate care according to patients’ individuality to improve self-management [42,43].

### Study Limitations

This study did not probe into the long-term causal relationship associated with the self-management of stroke patients. Thus, it would be beneficial to perform a rigorous effectiveness evaluation, and adopt experimental and longitudinal research designs when performing in-depth investigations on the COC and self-management of patients. Additionally, the enrollment site and inclusion of research participants in this study were limited to stroke patients in the neurology and neurosurgery ward of a medical center in central Taiwan. In this study, the self-management of the 4 patients diagnosed with their first stroke within 2–6 months was better than the other patients diagnosed with their first stroke within 1 month, which could only predict that the patients in the group of 2–6 months post-stroke had better self-management. Furthermore, although the stroke patients in this study were at the risk of disability from a mild stroke, these results could only represent the self-management status of the stroke patients in a medical center in central Taiwan. However, the extensibility of the overall research results means that this study may still serve as a reference for relevant units to provide appropriate care to patients. Moreover, this study used self-developed questionnaires on self-management, and the variable structure in the questionnaires might be insufficient; however, the reliability and validity of the scales may nonetheless be tested in the future to overcome this deficiency.

## 5. Conclusions

This study found that higher educational levels, exercise frequency of one to two times per week, experiencing a stroke within two to six months, and COC were predictors affecting the self-management of patients newly diagnosed with stroke. These variables were able to explain 44.5% of the total variance. Therefore, to improve stroke patients’ self-management, medical teams could help the patients to develop and strengthen their exercise habits and ensure COC, especially for those who have lower educational levels and those who have experienced a stroke within six months. Moreover, the government is advised to implement a COC model for case management in healthcare institutions in order to meet the individual care needs of patients affected by stroke. This model could provide diversified cross-team services, such as nursing, physical therapy, occupation therapy, nutrition, and psychotherapy services, to reduce the level of disability and improve physical independence and autonomy among patients. These management measures may, in effect, help patients to achieve the best self-management practices. Furthermore, the results of this study can serve as a guideline for medical teams in providing clinical care and improving self-management among patients affected by stroke.

## Figures and Tables

**Table 1 healthcare-09-00989-t001:** The basic demographic data and health status of the participants (*N* = 150).

	Variables	*n*	%
	Sociodemographic Characteristics		
Gender	Female	43	28.7
	Male	107	71.3
Marital status	Unmarried/widowed/divorced	37	24.7
	Married	113	75.3
Religious belief	No	133	88.7
	Yes	17	11.3
Smoking status	Non-smoker	71	41.3
	Smoker	79	58.7
Time affected by stroke	Within 1 month	146	97.3
	2–6 months	4	2.7
Educational level	Under elementary school	57	38.0
	Junior to senior high school	68	45.3
	College and above	25	16.7
Main source of income	Children/older siblings/spouse/parents	31	20.7
	Pension/government subsidy	43	28.7
	Work salary	76	50.7
Income	Somewhat insufficient/extremely insufficient	45	30
	Generally sufficient	99	66
	Sufficient with surplus	6	4.0
Frequency of exercise	Never	86	57.3
	1–2 times/week	25	16.7
	More than 3 times/week	39	26.0
Age	Mean (SD) = 65.15 (13.30)		
Total number of chronic diseases	Mean (SD) = 2.35 (1.35)		
ADLs IADLs	Mean (SD) = 87.10 (18.86) Mean (SD) = 19.85 (5.71)		
NIHSS	Mean (SD) = 1.89 (1.68)		
mRS	Mean (SD) = 1.82 (1.00)		

Note: ADLs, activities of daily living; IADLs, instrumental activities of daily living; NIHSS, National Institute of Health Stroke Scale; mRS, modified Rankin Scale.

**Table 2 healthcare-09-00989-t002:** The continuity of care and self-management of the participants (*N* = 150).

Variables	Mean	SD	Score Indicator
Continuity of care scale (12–60)	46.61	8.99	77.68
Relationships with providers in hospital (5–25)	19.46	3.82	77.84
Information transfer to patients (7–35)	27.14	5.35	77.54
Self-management (30–150)	110.50	15.12	73.67
Medical management (13–65)	51.74	6.80	79.60
Role management (12–60)	42.58	7.93	70.97
Emotional management (5–25)	16.18	3.33	64.72

**Table 3 healthcare-09-00989-t003:** Relationships among the basic demographic data, health status, and self-management of patients with stroke (*N* = 150).

Variables	Self-Management	Medical	Role	Emotional
Gender ^a^	1.226	0.481	1.071	1.920
Female				
Male				
Marital status ^a^	−0.194	−0.038	−0.249	−0.207
Unmarried/widowed/divorced				
Married				
Religious belief ^a^				
No	1.702	1.847	1.744	0.924
Yes				
Smoking status ^a^	0.734	0.593	0.553	0.800
Non-smoker Smoker				
Time affected by stroke ^a^	−2.560 *	−2.112 *	−2.311 *	−3.815
Within 1 month 2–6 months				
Educational level ^b^	17.152 ***	14.533 ***	20.303 ***	2.097
Under elementary school Junior to senior high school College and above				
Main source of income ^b^	3.422 *	3.016	5.306 **	0.376
Children/older sib-lings/spouse/parents Pension/government subsidy Work salary Scheffe post-comparison				
Income ^b^	1.352	1.612	2.847	2.491
Somewhat insufficient/extremely insufficient				
Generally sufficient				
Sufficient with surplus				
Frequency of exercise ^b^	3.883 *	3.022	5.093 *	1.429
Never 1–2 times/week More than 3 times/week Scheffe post-comparison				
Age ^c^	−0.359 **	−0.307 **	−0.393 **	−0.066
Total number of chronic diseases ^c^	−0.111	−0.057	−0.134	−0.069
ADLs ^c^ IADLs ^c^	0.273 ** 0.277 **	0.277 ** 0.274 **	0.287 ** 0.282 **	0.011 0.034
NIHSS ^c^	−0.181 *	−0.151	−0.222 **	−0.018
mRS ^c^	−0.323 **	−0.306 **	−0.347 **	−0.012

Note: ADLs, activities of daily living; IADLs, instrumental activities of daily living; NIHSS, National Institute of Health Stroke Scale; mRS, modified Rankin Scale. * *p* < 0.05, ** *p* < 0.01, and *** *p* < 0.001. ^a^ *t*-test, ^b^ *F*-test, and ^c^ *Pearson’s* product–moment correlation coefficient.

**Table 4 healthcare-09-00989-t004:** Relationships between the continuity of care and self-management of patients with stroke (*N* = 150).

Variables	Self-Management	Medical	Role	Emotional
Continuity of care scale	0.545 **	0.650 **	0.467 **	0.034
Relationships with providers in hospital	0.520 **	0.597 **	0.473 **	0.015
Information transfer to patients	0.544 **	0.666 **	0.447 **	0.046

Note: ** *p* < 0.01.

**Table 5 healthcare-09-00989-t005:** Important predictors affecting the self-management of patients with stroke (*N* = 150).

Variables	Self-Management						
	B	SE	Beta	Adjusted *R*^2^	*t*	95%CI	*p*
Educational level (reference group: under elementary school)	12.751	2.489	0.315	0.108	5.122	(7.831, 17.672)	0.001 ***
College and above							
Frequency of exercise (reference group: never exercising)	6.449	2.485	0.159	0.021	2.596	(1.539, 11.360)	0.01 **
1–2 times/week	16.548	5.737	0.177	0.024	2.884	(5.208, 27.888)	0.01 **
Time since stroke (reference group: within 1 month) 2–6 months	0.847	0.103	0.504	0.292	8.208	(0.643, 1.050)	0.001 ***
Continuity of care	12.751	2.489	0.315	0.108	5.122	(7.831, 17.672)	0.001 ***

Note: Linear regression was used for data analysis. B, unstandardized regression coefficient; adjusted *R*^2^ = 0.445, *F* = 30.921, ** *p* < 0.01, and *** *p* < 0.001.

## Data Availability

The datasets analyzed during the current study are available from the corresponding author upon reasonable request.

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
