# Peer review of "Continuity of Care and Self-Management among Patients with Stroke: A Cross-Sectional Study"

_healthcare, 2021, doi:10.3390/healthcare9080989_

Round 1

Reviewer 1 Report

The Authors present an interesting and original research about Continuity of Care and Self Management from patients suffering with recent Stroke. 

In my opinion, it's not necessary to give detailed explanation about NIHSS and mRS to readers which will probably be stroke physicians.

On the contrary, I do not find, maybe it was an attached document but I don't find it, the detailed of other scales namely the Chinese COC Questionnaire, the Stroke self-management scale. 

Finally, my major concern is about results presentation. The text is a little bit confusing and the tables are hardly understandable. This is obviously very important and must be improved before publication. 

Author Response

Dear Reviewer,

Thank you for your kind letter of 13 July 2021. We find your comments to be very important, and we believe that your suggestions and editing will greatly improve the quality of our paper. We have, hence, revised our manuscript as advised and have marked our corrections in red with underline in accordance with the reviewers’ comments. Additionally, we have carefully proofread the manuscript to minimize typographical, grammatical, and bibliographical errors.

Reviewer :

  1. In my opinion, it's not necessary to give detailed explanation about NIHSS and mRS to readers which will probably be stroke physicians.

On the contrary, I do not find, maybe it was an attached document but I don't find it, the detailed of other scales namely the Chinese COC Questionnaire, the Stroke self-management scale. 

Thank you very much for your suggestions.

As suggested, we have added detailed information about the Chinese COC Questionnaire and the Stroke Self-Management Scale (please see lines 140–153 on pages 3–4 and lines 161–190 on page 4). The detailed contents are listed in Annex 1.

  1. Finally, my major concern is about results presentation. The text is a little bit confusing and the tables are hardly understandable. This is obviously very important and must be improved before publication. 

We are thankful for your suggestions.

We have revised the typesetting of the four tables as follows: Table 1. The basic demographic data and health status of subjects (please see page 5-6); Table 2. The continuity of care and self-management of subjects (please see page 6); Table 3. Relationships among the basic demographic data, health status, and self-management of patients with stroke (please see page 7-8); and Table 4. Relationships between the continuity of care and self-management of patients with stroke (please see page 8).

In addition, the contents of the manuscript have been carefully typeset and edited to minimize grammatical and bibliographic errors.

Thank you and all of the reviewers for the kind advice.
Yours Sincerely,

Hsiao-Mei Chen, RN, PHD

E-mail: fiona@csmu.edu.tw

Reviewer 2 Report

Thank you for the opportunity to review this manuscript, which describes a cross sectional study examining a number of demographic, disease and care factors affecting self management of patients affected by stroke in Taiwan. While the research design is appropriate for the question, and the topic is of importance to ensure high levels of quality of care for stroke survivors, there were a number of issues with the manuscript that I believe need to be addressed.

  • The title is misleading – the authors do not examine factors affecting continuity of care.
  • Overall, I found the logic of the introduction difficult to follow. I was not convinced of the justification for this study based on the information presented. Self-management and interventions to improve self management in stroke patients are numerous (eg Fryer  CE, Luker  JA, McDonnell  MN, Hillier  SL. Self management programmes for quality of life in people with stroke. Cochrane Database of Systematic Reviews 2016, Issue 8; Sakakibara, Brodie M., Amy J. Kim, and Janice J. Eng. "A systematic review and meta-analysis on self-management for improving risk factor control in stroke patients." International journal of behavioral medicine1 (2017): 42-53). The introduction needs to clearly explain how this study advances what is already known in the context of so many self management interventions, and how this may improve care for people following stroke.
  • While the descriptions of the psychometrics of the measurement tools was helpful, additional explanation of why that specific tool for continuity of care measurement was chosen as opposed to others commonly used in the literature. Additionally, what was the justification for combining four measures of self-management? Also line 150-151 appears to have been left in by error as this in instructions.
  • Table 1 needs to be substantially edited to improve clarity. The formatting made it very difficult to understand the results, and “r” is not defined. The way the scheffe comparisons are presented could be removed and just presented in text as this was very complex. I found the means for the self management scale and subscales difficult to understand and meaningfully compare, as these don’t appear to be standardised to account for different numbers of items. Line 196- is it correct that the mean for medical self-management was higher than the self management mean score? How can this be when the emotional mean score was lower?
  • Line 210 – as this was not a study of causation, the authors cannot state that a variable “affected” or didn’t affect self-management – rather factors are only associated with/not associated with.
  • Line 261 – what do the authors mean by virtual variables?
  • Did the authors explore multicollinearity of the independent variables? I would hypothesise that education level may be correlated with continuity of care for example (eg Napolitano, Francesco, et al. "Assessment of continuity of care among patients with multiple chronic conditions in Italy." PloS one5 (2016): e0154940) but this does not appear to have been explored and accounted for in the regression.
  • I would encourage the authors to review the way the discussion is written. I’m not sure if is related to a language barrier, but to say that “patients over 65 years old probably believed their age would hinder them from changing their long term eating habits...” is not scientific, and not a valid interpretation of the data. All that can be concluded is that age was negatively correlated with self management – ie that younger patients scored higher on the measure of self management. There are a number of potential explanations for this, but these need to be presented as such, and not as fact. This issue is repeated several times throughout the discussion.
  • Based on the description of the sample (eg inclusion of only those with low mRS scores, low average NIHSS scores, High ADL scores) suggest this group of patients were minimally impaired by their stroke – how representative is this of all stroke survivors and what implications does this have for study generalisability?
  • The entire manuscript needs substantial editing from a native English speaker for conciseness, scientific expression and clarity.

Author Response

Manuscript Title: Continuity of Care and Self-management among Patients with Stroke: A Cross-Sectional Study

Date: 20 July 2021

Reply to the comments on Healthcare-1294221

Dear Reviewer,

Thank you for your kind letter of 13 July 2021. We find your comments to be very important, and we believe that your suggestions and editing will greatly improve the quality of our paper. We have, hence, revised our manuscript as advised and have marked our corrections in red with underline in accordance with the reviewers’ comments. Additionally, we have carefully proofread the manuscript to minimize typographical, grammatical, and bibliographical errors.

Thank you and all of the reviewers for the kind advice.
Yours Sincerely,

Hsiao-Mei Chen, RN, PHD

E-mail: fiona@csmu.edu.tw

Round 2

Reviewer 2 Report

I thank the authors for their thoughtful consideration of my comments and their comprehensive response. I think most of the issues have been addressed and the manuscript has improved as a result. However, I still have several reservations about the manuscript, which I have detailed below.

  • The authors have still not justified their choice of the Stroke self-management scale, and why they have adapted it based on the contents of 4 other surveys. Why were other scales inadequate or not appropriate for use in this study, and why was the authors self-developed survey better than other options?
  • Further to this point, given the Stroke self management and the continuity of care scales are now detailed in the appendix, I don’t think the individual items need to be detailed in the methods section – it makes it very difficult to follow. Just refer to the appendix.
  • The approach taken for the score standardization of the self management sub-scales needs to be explained in the methods (data analysis).
  • Participants should be referred to as such, not subjects. Participants is a more respectful term.
  • There are several places where "on the contrary" is not used appropriately.  I think further English language editing would improve the manuscript
  • Recommending that doctors encourage people who have had a stroke to participate in sports while in hospital (pg 10, line 361) does not seem feasible or appropriate. Physiotherapy however would be appropriate.not sure if this is an error in English translation?
  • Given only 4 participants were in the group 2-6 months post stroke, I would be very cautious about concluding anything about this group, particularly in terms of this predicting their self management. These individuals would appear to be a bit of a different group?

Author Response

Manuscript Title: Continuity of Care and Self-management among Patients with Stroke: A Cross-Sectional Study

Date: 28 July 2021

Reply to the comments on Healthcare-1294221

Dear  Reviewers,

Thank you for your kind letter of 25 July 2021. We find your comments very important, and we believe that your suggestions and editing will greatly improve the quality of our paper. We have, hence, revised our manuscript as advised and have marked our corrections in red with underline in accordance with the reviewers’ comments. Additionally, we have carefully proofread the manuscript to minimize typographical, grammatical, and bibliographical errors.
